# Trans-crustal structural control of $CO_2$-rich extensional magmatic systems revealed at Mount Erebus Antarctica

G. J. Hill [1,2✉], P. E. Wannamaker [3], V. Maris[3], J. A. Stodt[4], M. Kordy[3], M. J. Unsworth[5], P. A. Bedrosian[6], E. L. Wallin[7], D. F. Uhlmann [8,9], Y. Ogawa [10] & P. Kyle [11]

Erebus volcano, Antarctica, with its persistent phonolite lava lake, is a classic example of an evolved, $CO_2$-rich rift volcano. Seismic studies provide limited images of the magmatic system. Here we show using magnetotelluric data that a steep, melt-related conduit of low electrical resistivity originating in the upper mantle undergoes pronounced lateral re-orientation in the deep crust before reaching shallower magmatic storage and the summit lava lake. The lateral turn represents a structural fault-valve controlling episodic flow of magma and $CO_2$ vapour, which replenish and heat the high level phonolite differentiation zone. This magmatic valve lies within an inferred, east-west structural trend forming part of an accommodation zone across the southern termination of the Terror Rift, providing a dilatant magma pathway. Unlike $H_2O$-rich subduction arc volcanoes, $CO_2$-dominated Erebus geophysically shows continuous magmatic structure to shallow crustal depths of < 1 km, as the melt does not experience decompression-related volatile supersaturation and viscous stalling.

[1] University of Canterbury, Gateway Antarctica, Christchurch, New Zealand. [2] Institute of Geophysics, Czech Academy of Science, Prague, Czech Republic. [3] University of Utah, Energy & Geoscience Institute, Salt Lake City, UT, USA. [4] Numeric Resources LLC, Salt Lake City, UT, USA. [5] University of Alberta, Department of Physics, Edmonton, AB, Canada. [6] United States Geological Survey, Denver, CO, USA. [7] University of Hawaii at Manoa, Hawaii Institute of Geophysics and Planetology, Honolulu, HI, USA. [8] First Light Mountain Guides, Chamonix, France. [9] University of Lausanne, Department of Earth Science, Lausanne, Switzerland. [10] Tokyo Institute of Technology, Volcanic Fluid Research Centre, Tokyo, Japan. [11] New Mexico Institute of Mining and Technology, Socorro, NM, USA. ✉email: gjhill@ig.cas.cz

Alkalic volcanoes in rift settings stand in stark contrast to calc-alkaline volcanoes in arcs, such as the ring of fire around the Pacific Ocean. Arc magmas typically have volatiles dominated by $H_2O$ from dehydration of subducted oceanic crust. Basaltic alkalic magmas are $CO_2$-rich because they result from low degrees of partial melting of a mantle source often showing signs of being complexly metasomatized[1]. Rift settings and alkalic magmas contribute enormous amounts of $CO_2$ to the global exosphere budget, with climate-forcing level increases of atmospheric concentrations documented several times during Earth's history[2]. Phonolitic-series magma is alkali-rich and highly evolved from its parental basanite. Mount Erebus is the only phonolitic volcano with a persistent summit lava lake[3]. Compositionally uniform phonolitic lavas have dominated Erebus volcano eruptions for the last 17 ka[4] (Fig. 1). Phonolite magmas in other settings have been explosively destructive[5]. However, they also provide important resources such as rare earth elements and precious metals and thus are important to understand[6,7]. Erebus offers a unique opportunity to investigate magmatic system

processes from the mantle source to the surface and the enigmatic tectonic-structural controls of extensional magmatism and volatile release.

The alkali-rich Erebus phonolite has been modeled as evolving by fractional crystallization of a parental basanite sourced from an enriched upper mantle through a series of intermediate composition magma reservoirs[3,8,9]. Simplistic calculations suggest the formation of Erebus volcano requires >4000 km³ of basanite, which in turn requires low degrees of partial melting of >80,000 km³ of the upper mantle peridotite[3]. $CO_2$ emissions from the Erebus lava lake and fumaroles exceed the rates possible from estimated magmatic solubilities and crystallization, thus requiring deep-sourced, upper mantle $CO_2$ streaming through the plumbing system. This gas flux is an important heat supply, buffering the conditions of final phonolite differentiation that is modeled to be near 1000 °C in the 4–7.5 km depth range[9–12]. Its desiccating effect raises the solidus temperature of the magma promoting crystallization and differentiation. Despite intriguing insights from igneous petrology, no geophysical images have been

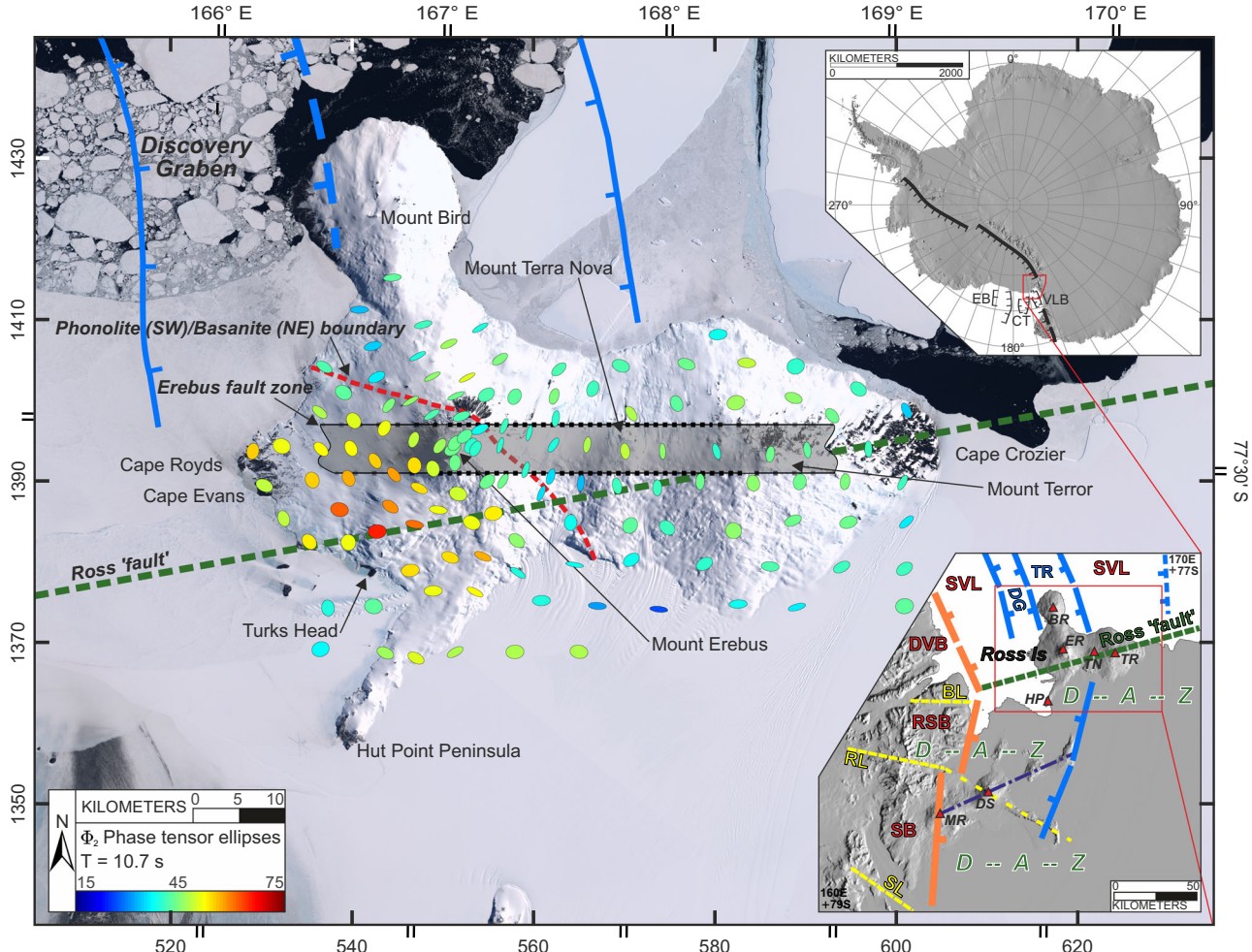

**Fig. 1 Structural setting of the Mount Erebus magmatic system.** Site map showing impedance phase-tensor ellipses at the MT sounding locations over Mount Erebus and Ross Island during three field seasons; see text for color and shape designations. Coordinate ticks denote both UTM Zone 58S (singles) and latitude–longitude (doubles). Note the warm-colored ellipse region southwest of Mount Erebus indicating conductive volumes in the 0–7 km depth range approximately (Supplementary Information). Hypothesized Erebus fault zone contains principal low-resistivity structure resolved through the crust. The lower right inset shows regional structural setting[15–17], including the Ross Fault (green) in the accommodation zone of the terminus of the Terror Rift (TR) and Discovery graben (DG). Other symbols include Southern Victoria Land basin (SVL) and Discovery accommodation zone (D--A--Z). Yellow dashed lines represent other antithetic shear zones separating Dry Valleys block (DVB), Royal Society block (RSB), and Skelton block (SB). Volcano mounts include Bird (BR), Terror (TR), Terra Nova (TN), Erebus (ER), Hut Point (HP) Morning (MR), and Discovery (DS). Upper right inset places location of lower right view within Antarctica. The upper right orientation is ~180° rotated from the main and lower panels.

obtained to date which reveal the source-to-eruption magmatic process including external controls on pathways and staging.

Ross Island lies at the southern end of the late Miocene to Recent Terror Rift (Fig. 1), which is located within the larger and older Victoria Land Basin[13,14]. The Terror Rift terminates to the south across a broad region called the Discovery accommodation zone (Fig. 1) via a series of high-angle, antithetic shears[14–16] trending roughly E–W. Often included in these is the inferred Ross fault, conjectured based on aeromagnetic data[15,17]. Globally, accommodation zones are strongly associated with magmatism and alteration, and both may be influenced by pre-existing structures[18,19]. Erebus volcano dominates Ross Island and is radially surrounded at roughly 120 degrees by volcanic systems at Mount Bird and Mount Terror and on the Hut Point peninsula[3]. These visible edifices on Ross Island clearly demonstrate the late Miocene-present, Phase II basanite-phonolite trend of the broader Erebus volcanic province that extends into Victoria Land[20]. Erebus started forming about 1.3 Ma and is still active with many recent eruptions of lava flows around and to the southwest of the summit[21,22]. The magmatic evolution of the Erebus lineage contrasts with rocks sampled by the Dry Valley Drilling Project (DVDP) at the surrounding centers[3,8]. Olivine-hosted melt inclusions show that all magmas on Ross Island were very $CO_2$-rich. However, the DVDP eruptions were short-lived and possessed greater water contents. No primitive mantle-derived basanites have been reported at Erebus itself, but those from the surrounding volcanic centers are considered to represent its parental magmas[3,8].

The lack of deep-seated magmatic seismicity and crustal-scale earthquakes on Ross Island area reduces the resolution of seismic exploration of the volcanic system at all but the shallowest levels. Available sources for seismic imaging of Mount Erebus summit area include Strombolian eruptions, icequakes, very long-period gas slug ascent signals, surface and body wave interferometric effects, and shallow artificial explosions[23–25]. These seismic results indicate a complex velocity distribution in the uppermost few kilometers potentially corresponding to multiple intrusions and alteration zones. A common feature of the seismic models is a low-velocity channel dipping west-northwest for a few kilometers from the summit crater that appears to contain the source region for very long-period seismic signals[26]. However, seismic survey limitations leave a large gap in imaging, from a few kilometers depth in the crust through to the uppermost mantle. Global teleseismic tomography provides velocity models at scales tens of km or greater[27–29].

We are able to achieve a whole-system description of an active, $CO_2$-dominated rift volcano using the magnetotelluric (MT) geophysical technique. The MT data interpreted here provide structural resolution superior to seismic studies to date by imaging the electrical resistivity of the Erebus system continuously from the near-surface to a depth of tens of kilometers. This is possible due to the very broad bandwidth of the MT signals (~0.005–900 s wave period) and the relatively dense site spacing (1–5 km) covering the entirety of Ross Island. Polar regions such as Ross Island present additional challenges for MT data collection and processing[30] (i.e., high-impedance electrical contact, non-plane wave source effects). A rigorous approach[31] is applied to ensure high fidelity MT data and models ("Methods", Supplementary Information and Supplementary Figs. S1–S4). Our study identifies the magmatic system beneath Erebus volcano from its upper mantle source through the crust. We argue that dynamic intersection of the Terror rift and accommodation zone fault structure may control the geometry and style of magma transport and storage in the Mount Erebus system. We also show that this $CO_2$-dominated volcano possesses a unique geophysical structure compared to $H_2O$-rich subduction arc andesitic systems due to their fundamentally different chemical physics of volatile exsolution.

## Results

**Phase-tensor analysis**. Even before formal data inversion is undertaken, first-order features of the internal structure of Mount Erebus can be seen in the measured MT data, such as the impedance phase tensor[32–34] ($\Phi$) ("Methods" and Supplementary Information). The quantity $\Phi$ has the advantage that it can distinguish local changes in the shallow geology from deeper structural zones (e.g., static distortion). $\Phi$ can be defined graphically by an ellipse, where ellipse fill color represents a quantity termed ($\Phi_2$) the geometric mean of maximum and minimum phase that is not dependent on the coordinate system. The ellipticity represents the way that the data vary with azimuth i.e., is data 1D or 2D/3D. A value of $\Phi_2 > 45°$ (denoted by warm fill color) indicates decreasing resistivity with depth, while $\Phi_2 < 45°$ (cool fill color) implies resistivity increasing with depth, for the depth range sensed by the periods in question[35].

The observed $\Phi$ ellipse response map over Ross Island at a mid-range period of 10.7 s (Fig. 1 and Supplementary Fig. S5) shows warm colors (high values) southwest of Erebus summit extending towards Capes Royds and Evans to the west and Turks Head to the south. The area of warm ellipses (Fig. 1) lies within the sector of younger phonolitic vents and flows noted previously southwest of Erebus crater[21] corresponds with a general drop in apparent resistivities in that area (Supplementary Fig. S6). The cooler colors of $\Phi_2$ north and east of Erebus summit and over central and eastern Ross Island imply higher resistivity with depth, suggesting a lack of crustal-scale magmatic activity, while orientations are responding to large-scale resistivity structural grain. The observed vertical magnetic field transfer function or induction arrows $\mathbf{K}$ where $H_z = -\mathbf{K} \cdot \mathbf{H}$ point toward regions of lower resistivity ("Methods" and Supplementary Information). They come in two main groupings (Supplementary Fig. S7), one local to and pointing toward the summit crater, and the other clustered around the island shore and pointing toward the seawater and other possible offshore structures.

**Inverse modeling**. The resistivity model of Ross Island and Mount Erebus obtained by 3D nonlinear inversion explicitly includes topography and bathymetry (Supplementary Fig. S8) The adequacy of the mesh discretization was demonstrated through convergence calculations and the physical behavior of refracted EM waves[36]. The starting model for the inversion was a uniform 100 Ωm half-space. This value is a typical bulk crustal resistivity for active terrains in our experience, and also is consistent with volatile-free upper mantle peridotite at average thermal conditions[37]. The starting model included the surrounding bathymetry where seawater resistivity of 0.3 Ωm was imposed[38]. The phase ellipse, invariant apparent resistivity and induction arrow responses of the starting model (Supplementary Figs. S9–S11) highlight the need to include topography and bathymetry in the inversion. In particular, the Erebus crater edifice induces a pronounced drop in apparent resistivity that could be misinterpreted as anomalous volcanic structure if the topographic effect was not included[36,39]. The cluster of induction arrows local to and pointing toward the summit crater appears explainable largely by the topography alone.

The final models we show emphasize inversion of the four complex elements of the impedance tensor $\mathbf{Z}$ defined as $\mathbf{E} = \mathbf{ZH}$ ("Methods" and Supplementary Information). A 5% impedance element floor was applied (see Supplementary Information). We could not define good constraints on possibly thick sedimentary sections offshore[16,40] and the MT vertical magnetic field transfer

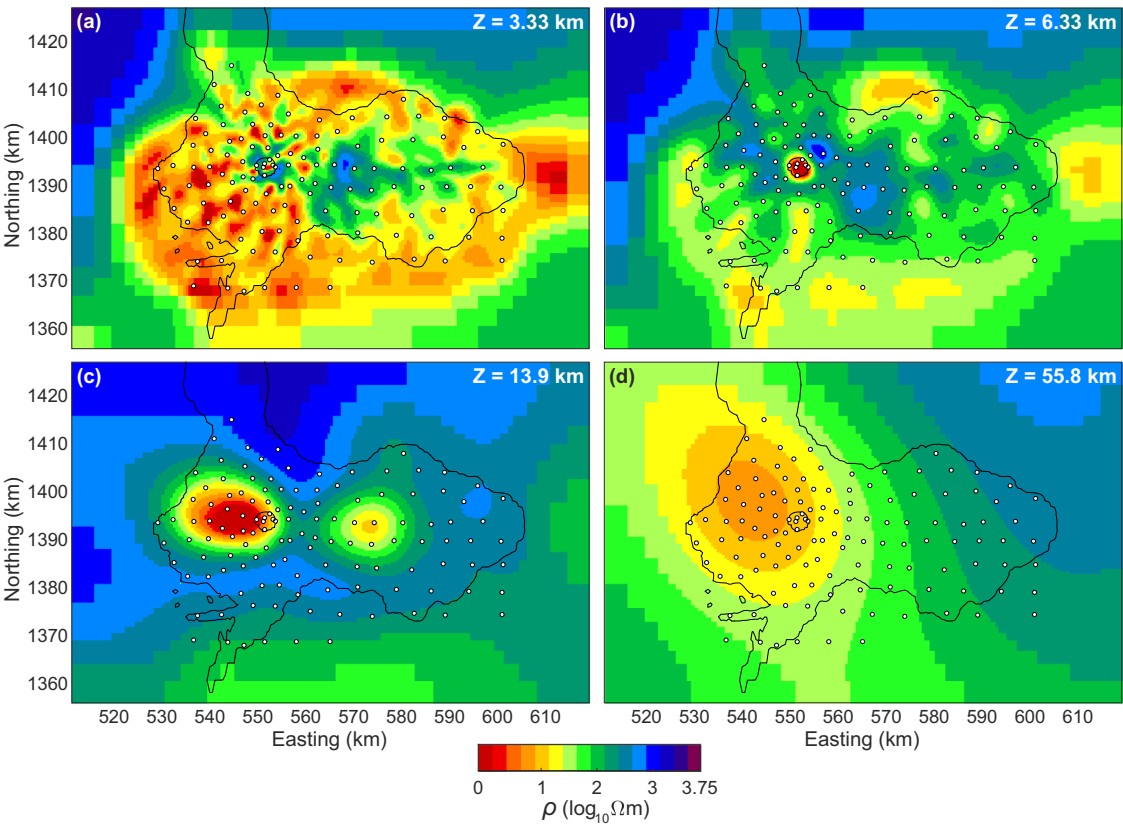

**Fig. 2 Map view resistivity sections.** Plan view model resistivity (ρ) slices at 3.3, 6.3, 13.9, and 56 km depths below sea level (**a–d**) under Mount Erebus and Ross Island from 3D MT finite element inversion, MT measurement locations are marked by white circles. Inversion testing suggests that low resistivity representing deeper magmatic input from the upper mantle likely rises steeply under western Ross Island although precise depth extent is challenging to bracket. A small, almost circular feature near Northing 1395 and Easting 550 is outline of the Erebus summit plateau. An alternate color map version is provided as Supplementary Fig. S24.

function is comprised entirely of secondary vertical magnetic field variations sensitive to lateral changes ("Methods" and Supplementary Information). The vertical magnetic field transfer function data hence were down-weighted with an error floor of 0.075 and restricted to periods less than 100 s in separate trial inversions. The impedance inversion exhibited monotonic convergence (Supplementary Fig. S12) and achieved an overall good normalized root-mean-square (nRMS) fit of 1.29 (Supplementary Fig. S13). The computed phase ellipses and apparent resistivities of this model compare closely to the observations (Supplementary Figs. S14 and S15). Example sounding curves west and east of Erebus summit crater are shown together with inversion model computed responses in Supplementary Fig. S16. The vertical magnetic field response is broadly consistent with the presented model structure (Supplementary Information and Supplementary Fig. S17). Inversion modeling using the vertical magnetic field transfer function in a constrained fashion yields a similar structure (Supplementary Figs. S18 and S19).

The 3D inverse resistivity model is displayed in a series of slices in both map and section views (Figs. 2 and 3). Spanning Ross Island and its near offshore areas in the upper 3–5 km is a rather clumped, low-resistivity layer beneath an irregular resistive thin quasi-layer (Figs. 2a, b and 3). That sequence is consistent with being a low-temperature clay alteration blanket that is typical of volcanic systems[41–43], overlain by the youngest volcanic rocks, which are not altered significantly due to low temperatures and lack of a well-developed groundwater/hydrothermal system. This blanket is thickest in the southwest sector of Ross Island where the young lava flows are concentrated (Fig. 3) and explains the concentration there of phase ellipse values exceeding 45° (warmer

colors) at 10.7 s and nearby periods (Fig. 1 and Supplementary Fig. S20). Oxygen isotope trends in the phonolitic lavas can be explained by limited incorporation of this hydrothermally altered rock[9]. We do not discuss this zone further.

The most striking and important feature of the 3D model is an elongate low-resistivity volume shown projecting from upper mantle depths, extending through the crust, and reaching Erebus summit (Figs. 2 and 3). The low-resistivity trend changes abruptly from being steeply upward, to possessing a shallow dip from 15–20 km depth in the lower crust and heading eastward. This low-resistivity segment continues to beneath the active crater, where it again turns steep and narrows toward the surface. The most visible impact of this conductor on the measured data is to induce warmer-colored phase ellipses of high ellipticity under the west-central portion of Ross Island at longer periods (Supplementary Fig. S21). A secondary compact and less conductive body lies between Mounts Terra Nova and Terror east of the main low-resistivity conduit (Figs. 2 and 3), but does not appear to extend to depth. Resistive middle to deep crust and uppermost mantle characterizes the eastern extent of Ross Island beneath the older Mount Terror area (Fig. 3).

**Model testing.** Beyond the prior tests on crustal contributions to the observed MT response, it is important to assess the requirement that low-resistivity structure extends into upper mantle depths where the basanite parental magma is generated[3]. To do this, we have run a constrained inversion (Fig. 4) where the model is clamped at the starting guess of 100 Ωm below the Ross Island Moho[27,28] (~23 km depth). The inversion test differs from the

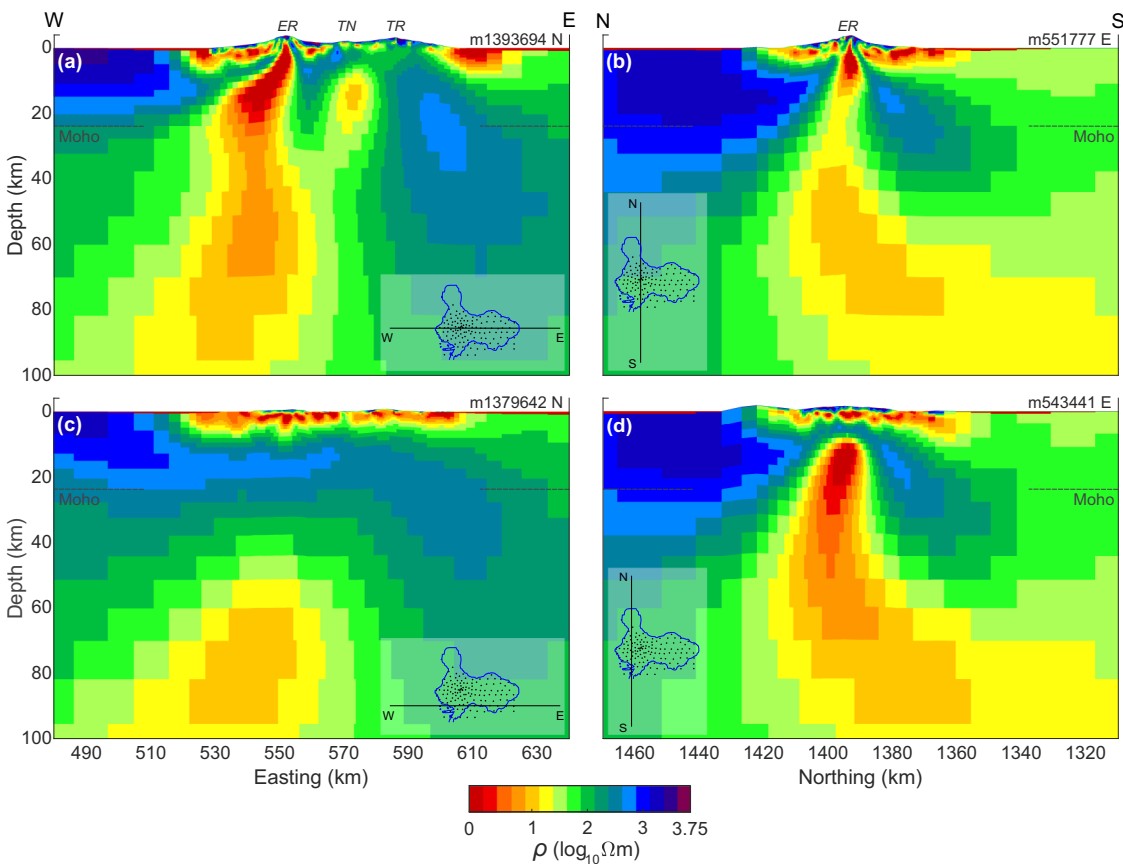

**Fig. 3 Vertical-section view of resistivity structure.** E–W (**a**, **c**) and N–S (**b**, **d**) model resistivity (ρ) section views across Mt Erebus and Ross Island from 3D MT finite element inversion. The top two panels (**a**, **b**) pass through the active summit crater. The structural control slanting the main interpreted magma pathway to the west with depth from the summit is clear in the upper left panel (**a**). At greater distances south panel (**c**), the conductor tends to move out of the page toward the reader and downward as implied by panel (**d**). The dashed horizontal line segments at edges of sections (**b**) and (**d**) represent the seismic Moho[27,28]. The three volcanic summits are labeled Erebus (ER), Terra Nova (TN), and Terror (TR) in panels **a** and **b**. An alternate color map version is provided as Supplementary Fig. S25.

simpler forward runs (Supplementary Information) in that the constrained inversion allows the additional structure to form elsewhere in the crust in an effort to compensate for the prevention of structure in the upper mantle. The inversion converged in 18 iterations to a nRMS minimum of 1.33, which is a modest increase from that of the preferred model at 1.29. Systematics to the nRMS increase are subtle although we suggest a slight preponderance of increases west of the crater (Supplementary Fig. S22).

The preferred unconstrained inversion model is compared with the Moho constrained model in W–E and N–S sections (Fig. 4). As expected, the constrained inversion pushes additional conductivity (low resistivity) vertically into the lower crust from the upper mantle where it was prevented from forming in order to replicate the response. The lowest resistivities fall to as small as 0.3 Ωm in the core of the main conductor at depths around 10 km; a value barely reached in the unconstrained inversion, it is significantly more prominent in the constrained. In addition, the constrained inversion shows higher resistivities flanking the main central conductor so that there also is an overall increase in resistivity contrast relative to the unconstrained inversion. Hence, more extreme model characteristics are required in the constrained model to achieve a similar data fit.

To investigate further, we have taken the computed response of the constrained inversion models (Fig. 4b, e), and added Gaussian noise to each response value equal to each data error in the field observation set. Gaussian noise was computed using the routines Ran2 and Gasdev[44]. The synthetic dataset was then inverted using the same inversion parameters as the preferred model (Figs. 2 and 3). If the extension of significant low-resistivity structure into the upper mantle in the preferred model is merely the result of regularization (smoothing), then a similar feature should form in this synthetic inversion even though the causative response structure has been confined to the crust. The model formed after 18 iterations from the synthetic, noise-added data appears in Fig. 4c, f. The nRMS achieved was 0.93, which is lower than values achieved for the natural dataset due to modest non-idealities in the latter (e.g., occasional data scatter apparently greater than the nominal error values).

Nevertheless, the recovered model low resistivity resides almost entirely in the crust (Fig. 4c, f) with little extension below the Moho. The slight extension that does appear continues to the southwest and does not become vertical, which is as expected with no information to the contrary. Hence, while the precise characteristic of the observations leading to the upper mantle low resistivity in the model (Figs. 2, 3, and 5) is not obvious, this test inversion implies that extension vertically downward west of the summit crater is required. Additional volumetric renderings (Supplementary Fig. S23) further illustrate the orientation of the westward dipping, high conductivity zone.

## Discussion
In light of the low resistivity expected for melts, it is compelling to associate the rising volume of low resistivity from west of Mount

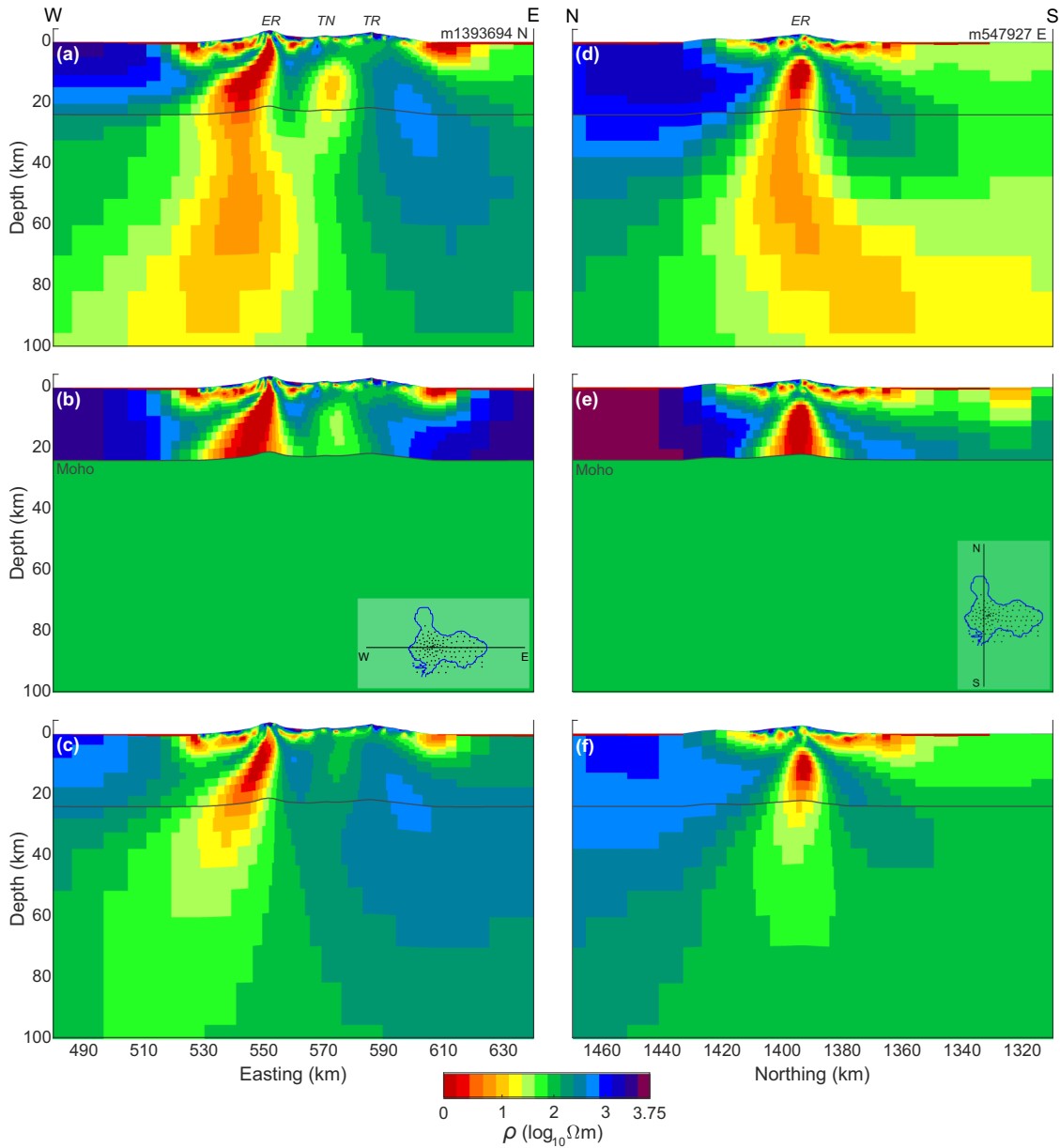

**Fig. 4 Resistivity model testing. a** E–W slice through Erebus summit of preferred inversion resistivity (ρ) model of Figs. 2 and 3; **b** E–W slice through Erebus summit of inversion model with all structure constrained to lie above the Moho[27,28]; **c** E–W slice through Erebus summit of synthetic inversion model derived from constrained inversion model response; **d–f** N–S slices analogous to the E–W views of (**a–c**). The three volcanic summits are labeled Erebus (ER), Terra Nova (TN), and Terror (TR) on panels **a** and **d**. An alternate color map version is provided as Supplementary Fig. S26.

Erebus with the main conduit delivering magma and $CO_2$ vapor upward[45]. The steep upper mantle portion of the conductor lies approximately in the southward projection of the eastern bounding normal fault of the Discovery half-graben[14] (Fig. 1), which shows >4 km of offset through the Miocene to the present. It is hence reasonable to correlate the concentrated active extension with upper mantle upwelling and partial fusion. Lab studies[46] have shown that phono-tephrite magma at a plausible temperature of 1250 °C has a resistivity of 1–0.3 Ωm for water contents of 0–3 wt. %. The nominal lowest resistivity in our inversion model at depths of 40–50 km is ~3 Ωm although uncertainty is substantial (Supplementary Information). Nevertheless, the deeper aspects of our model could be explained by a few 10 s % melt by volume.

For crustal levels, the resistivity of dry phonolite melt at 1000 °C, the interpreted temperature of the magma storage zone in the 4–7.5 km depth range[9–11], is ~1 Ωm[46]. At somewhat higher

temperatures with limited water contents that may pertain to greater crustal depths[10,11], resistivities[46] could be as low as 0.3 Ωm for 1050 °C, 1.1 wt % $H_2O$. This would suggest almost pure liquid melt in the core of the inverted resistivity structure (Figs. 2 and 3), although exact melt fraction estimates using crystal-melt mixing models[45] may not be justified given the effect of model regularization. In addition, a change in melt temperature by a mere 50 °C may lead to melt resistivity[46] changing by a factor of at least 1.5. Dry $CO_2$ is a nonpolar molecule, which in principle should not carry significant solute[47] and thus not be conductive. However, speculatively there may be yet unrecognized minority volatile phases that could contribute to the conductivity thereby reducing the melt fraction needed to explain the observed resistivity anomaly.

It also seems inescapable that the abrupt turn in magma/fluid trajectory in the deep crust from being steep below the Moho, to

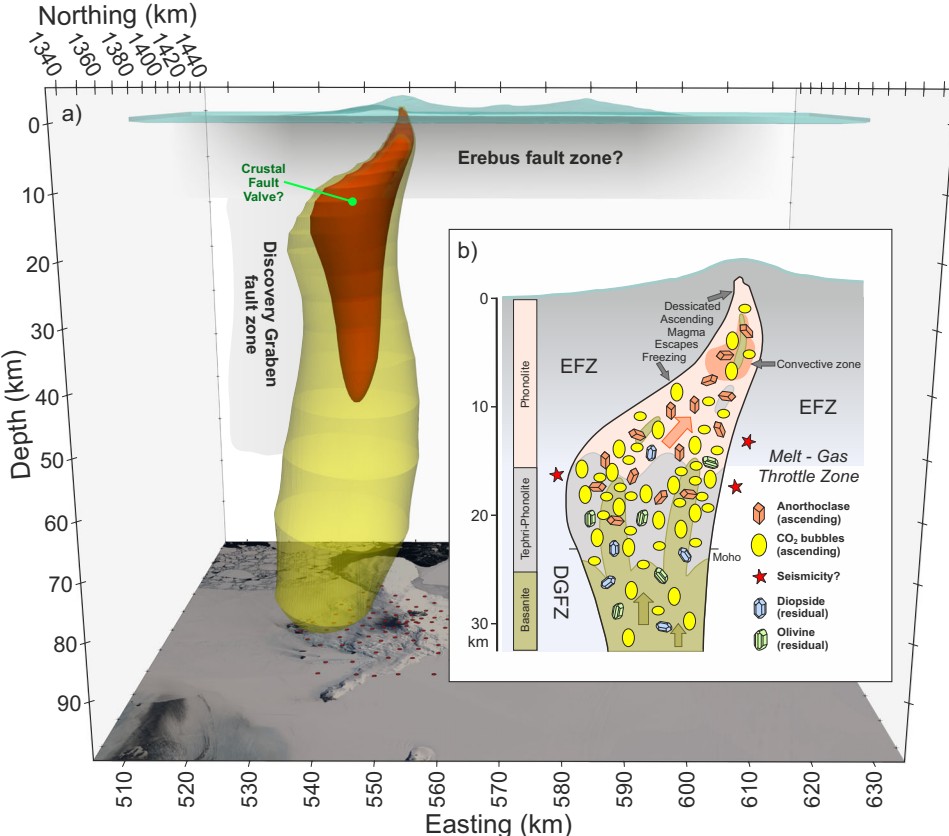

**Fig. 5 3D resistivity model and magmatic processes. a** 3D visualization facing Mercator north of the 5 (red) and 10 (yellow) Ωm resistivity contours from the 3D finite element inversion. Gray band extending to 15 km depth is the conjectured Erebus fault zone (EFZ) within which the crustal Erebus conductor and possibly the minor Terra Nova conductor lie. Also shown is the projection of the deepest offset portion of the Discovery graben fault zone[14]; **b** schematic depiction of magmatic processes conjectured to be occurring nominally in the core zone of the MT low resistivity, generally following recent petrological results[2,4]. Nominally, two channels of upward flow concentration from a deep-middle crustal throttle zone exemplify episodic breakthrough of $CO_2$ and entrained magma. The convective melt zone drawn in the 4–8 km depth range[11] is compatible with the inversion cross-section of Fig. 3b. Spatially continuous upflow of $CO_2$-dominated magma is in contrast to depth-limited magma zones of $H_2O$ arc volcanoes.

being east-moving upward to the summit, is structurally controlled (Fig. 5a). We argue that this sharp turn represents a sort of choke-point or bottleneck where magma evolution and degassing may be promoted (Fig. 5). It is at the appropriate depth to represent the inferred "nozzle"[10] across which periodic injections of parental basanite and less evolved tephrophonolite together with mantle-sourced $CO_2$ appear to take place. Melt inclusions within anor-thoclase crystals in expelled phonolitic lava bombs suggest incor-poration depths approaching 15 km[10], which lie toward the deeper reaches of our moderately west-dipping mid-crustal conductor.

The inferred magmatic/volatile cycling, together with the implication from the low-resistivity geometry that the oblique Terror Rift and accommodation zone structures may intersect in the deep crust, are strongly suggestive of fault-valve behavior[48,49]. In this phenomenon, externally sourced fluids from depth build in pressure near the mid-crustal, brittle-to-ductile rheological transition. Sufficient pressure increases may trigger dilational rock rupture or shear activation of favorably oriented planes of pre-existing weakness[48,50]. Such structural failure, in concept, temporarily extends embrittlement downward into otherwise ductile material tapping pressurized fluids or, in this case, $CO_2$ vapor and melt. Upward channelized escape of $CO_2$ could entrain deeper less evolved magma such as evident in the melt inclusions[10] (Fig. 5b). This process may repeat if the vapor supply continues for a prolonged period. Unless differential stresses are already great, the requisite fluid pressures will be a significant fraction of lithostatic, and well above hydrostatic, across the range

of extensional to compressional stress regimes[49]. The build-up of magma and $CO_2$ originating in the upper mantle plausibly occurs at the deep crustal transition from relatively ductile behavior below to brittle behavior above the sharp turn in the major conductor. Seismic activity since first recorded in 1973 has been dominated by Strombolian eruptions and infrequent tremor, with no crustal earthquakes reported[51], but seismic array coverage has been limited.

That pulsed mantle $CO_2$ streaming may reflect opening of crustal fractures to release deep fluid overpressures has been suggested previously[10]. The E–W trend of our main crustal conductor, which we portray as lying in a hypothetical Erebus fault zone (Figs. 1 and 5), is somewhat oblique to, but not strongly disoriented from, the interpreted regional Ross fault. We hence suggest that accommodation zone structures effecting southward termination of the Terror Rift may control the geometry of the pathway transporting Erebus magmas and $CO_2$ vapor toward the surface. The portrayed Erebus fault zone closely parallels the trend of the summits of Erebus, Terra Nova and Terror, which were proposed to occupy a large-scale, E–W fissure generated by upward mantle plume pressure centered on Erebus[3,8]. That fis-sure was not postulated to extend west of Mount Erebus, nor do we see low-resistivity trends projecting toward Mount Bird or Hut Point peninsula in any companion fissures, but some form of E–W structural weakness along Ross Island is still proposed.

The probable extension of the main conductor into the upper mantle is comparable to the petrological source region of the

parental basanite magmas plus deep differentiation with upwelling to tephri-phonolitic compositions[10] (Fig. 5b). This seems the likely pathway also for the steady $CO_2$ streaming from the upper mantle that provides the principal heat source and controlling volatile environment for the crustal phonolite differentiation. Mount Erebus volcano is unique in the Southern Victoria Land region in possessing a sustained connection of upper mantle parental magma and $CO_2$ gas, whereas the surrounding magmatic vents Hut Point, Dry Valleys, Mount Terror, and Mount Bird districts are characterized by higher water content and were short-lived with little heat/melt resupply[8]. Deep extension and its complex termination associated with the southern end of the Discovery graben may be required to make space for buoyant ascent of magma and gas through a region of ductile lithology from the source region of $CO_2$-rich basanite melt. The origin of carbon dioxide in volcanic regions is typically ascribed to previously subducted carbonates or to plumes[52], although the depth and age of the source for Ross Island and Victoria Land are debated[53,54].

It is significant that the low-resistivity zone beneath Erebus volcano extends continuously upward to within ~1 km of the surface under the crater (Figs. 3 and 5a). This illustrates a fundamental difference between $CO_2$-dominated volcanoes and the more familiar $H_2O$-dominated subduction arc systems. Low-resistivity zones under arc volcanoes are typically confined to depths greater than 5 km, as exemplified by Mounts Ruapehu[55], St Helens[35,39,56], Tongariro[43,57], Naruko[42], and Azumayama[58]. Coincident seismic low velocity or high attenuation structures in the middle crust have been widely reported[58–62]. The depth of magma bodies beneath subduction zone volcanoes may be controlled by the amount of $H_2O$ that is dissolved in the magma[63,64], although rheology could play a role[65]. Magma bodies beneath subduction arc volcanoes at such depths are water-saturated at ~4 wt % $H_2O$, which in turn is the water content typically inferred from crystal-melt inclusions in arc eruptive products[63,64]. Because the water solubility of such magmas decreases as pressure decreases, upward motion of magma can lead to supersaturation, water exsolution and viscous stalling[63,64].

Phonolitic magmas at similar depths certainly are capable of carrying a similar content of dissolved water[6]. However, the upper crustal phonolite magma at Mount Erebus, as inferred from crystal-melt inclusions, is very dry, on the order of 0.1 wt % $H_2O$[10]. This water content would result in saturation only within a few hundred meters of the surface[6,63,64]. The dry state at shallow depths is believed to be the result of magma dessication via streaming of the dominant mantle-sourced $CO_2$ vapor[10]. The amount of $CO_2$ dissolved in the present phonolite magma is relatively small, <1 wt %, which degasses mostly passively in the Erebus crater lava lake[10]. As a result, the upwelling Erebus phonolite magma does not stall through volatile exsolution as observed in arc volcanos. This results in the continuous magmatic system that is clearly imaged by MT data as a low-resistivity pathway extending from the upper mantle to the summit crater of Mount Erebus. The lack of volatile exsolution-related stalling may in part explain the unusually long-standing, open conduit/lava lake conditions that characterize the Mount Erebus system. The primary tectono-structural fault-valve dynamic of the Erebus magmatic system represents an inherited tectonic control that regulates magma movement to the upper crust and $CO_2$ flow to Earth's atmosphere. Further data collection to increase survey aperture to the southwest could provide improved resolution of the deeper magmatic system in the upper mantle.

## Methods

**The magnetotelluric method**. MT exploits natural electromagnetic (EM) wavefields, generated by global lightning and ionospheric electrical current sources, to resolve Earth resistivity (sometimes referred to by its inverse, conductivity) structure. The broadband nature of the sources (0.002 to >1000 s wave period, or 500 to <0.001 Hertz frequency) allows resolution from ~100 m depth (high-frequency or short-period data) to the upper mantle (low-frequency or long-period data)[66]. Resistivity is sensitive to small amounts of fluids (including melts)[45] and has yielded insightful images of shield volcanic systems, revealing likely source zones, pathways, and inherited controlling structures. However, to date, this method has been applied mainly in subduction arc regimes[35,42,43,55–57,67,68].

**Magnetotelluric survey**. During three-austral summer field campaigns from 2014 to 2017, MT measurements were completed mainly via helicopter deployment yielding 129 high-quality sites over Mount Erebus and the greater Ross Island region (Fig. 1). Station occupation times varied from a minimum of 3 days to a maximum of 12 days. Station spacing was 1.5–2 km surrounding the Erebus edifice and increased to ~5 km for distal sites on the McMurdo and Ross ice shelves. To overcome the high contact impedance of polar ice sheets when measuring the electric field component, custom buffer preamplifiers were deployed at each bipole electrode (Supplementary Information and Supplementary Fig. S1), as is now standard[30,31,69–72].

High-quality MT tensor impedance (**Z**) and vertical magnetic field (**K**, induction arrow) transfer functions generally were obtained for the period band of 0.004–1000 s through robust remote reference outlier removal[66], using both magnetic and electric reference fields. Nonplanar EM wave components that did not meet the MT assumption were either negligible or successfully removed, as verified by comparing MT response functions during high and low diurnal signal intervals[30,31,72] (Supplementary Information, Supplementary Figs. S2–S4). Data at each station were collected in geomagnetic co-ordinates, and then rotated to a coordinate system having the x-axis aligned with Mercator north as shown in Fig. 1 for modeling.

**Phase-tensor analysis and inversion**. The electric field recorded in MT measurements is commonly distorted through near-surface heterogeneity (e.g., complex surficial geology including alteration zones, and different volcanic units/structure) of the resistivity structure[33]. The phase tensor **Φ**, preserves the underlying regional phase information free from distortion[32,34]. The phase relationship intrinsic to the impedance tensor **Z** forms a second rank 2D tensor[32,34] defined by the matrix equation $\mathbf{\Phi} = \mathbf{X}^{-1}\mathbf{Y}$ where $\mathbf{X} = \mathrm{Re}(\mathbf{Z})$ and $\mathbf{Y} = \mathrm{Im}(\mathbf{Z})$. **Φ** is an observational indication of the inherent large-scale variations in resistivity structure[32–35], illustrated graphically as an ellipse with the greatest and least phase corresponding to the principal axes of the ellipse.

Because the propagation of EM wavefields in the Earth at typical periods is diffusive, and measured data are finite in number and contain errors, constructing a stable 3D resistivity model of the Erebus magmatic system requires stabilized nonlinear inversion. To create this we apply the HexMT finite element algorithm[36,39] which can accommodate the steep topography of Ross Island and Mount Erebus in the MT response through vertical deformation of its hexahedral elements. Stabilization (regularization) of the isotropic resistivity model was achieved using the common approach of damping (smoothing) of the model. Diffusive propagation also means that signals attenuate downward according to frequency; thus, structural resolving scales broaden with depth. Model robustness and feature necessity were verified through resolution testing (Supplementary Information), where features are perturbed and resultant change in data misfit or model plausibility assessed.

## Data availability

The observed MT dataset in community standard edi format has been uploaded to the USAP-DC repository with https://doi.org/10.15784/601493.

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

## Acknowledgements

Logistical support was provided by Antarctic New Zealand and the United States Antarctic Program including flight support from Southern Lakes Helicopters—S. Clarke, M. Deaker, M. Hayes, A. Hefford, J. Laing, S. Mullally; and Petroleum Helicopters Inc.—H. Blake, K. Cox, S. Firth, J. Heaslet, M. Jansen, J. Radford, R. Skoreki, & B. Wilson. Mountaineering support from T. Arnold, P. Biskind, R. Bottomley, A. Christophers, R. Hunter, M. Kingsbury, & B. Nicholson; and field assistance from K. Allen, E. Buys, M. Cloutier, K. Grenfell, B. Goodsell, A. Hindle, C. Lodge, & S. Parker is appreciated. Base maps and digital elevation models were provided by M. Cloutier at the Polar Geospatial Center, University of Minnesota. Discussions with S. Bannister and H. Bibby of GNS Science improved the clarity of the manuscript. Alternate color map versions of Figs. 2–4 are provided as Supplementary Figs. S24–S26. The authors suggest the use of the Visolve (https://www.ryobi.co.jp/products/visolve/en/) online utility by readers interested in alternate color scales for other figures. Any use of trade, firm, or product names is for descriptive purposes and does not imply endorsement by the U.S. Government. New Zealand Royal Society Marsden award: ASL-1301 to G.H. Lumina Quaeruntur Fellowship: LQ100121901 to G.H. US National Science Foundation Office of Polar Programs award: AES-1443522 to P.W. US Department of Energy Geothermal Technologies Office contract: DE-EE0002750 to P.W. Natural Science and Engineering Research Council Discovery Grant to M.U. US National Science Foundation Office of Polar Programs awards ANT1141534, PLR1644234 to P.K.

## Author contributions

G.H. and P.W. designed the research program with assistance from P.K. G.H., P.W., V.M., J.S., M.K., M.U., P.B., E.W., and D.U. conducted the data collection. D.U. and G.H. developed the field operations and safety plan. G.H. carried out the time series and phase-tensor analysis. V.M., P.W., and M.K. computed the inverse models. G.H., V.M., and P.W. constructed the figures. J.S. designed and built the electric field measurement buffer amplifiers. G.H. and P.W. wrote the initial draft. All authors contributed to the interpretation and preparation of the final manuscript.

## Competing interests

The authors declare no competing interests.
