## [Peer Review File · Nature Communications]

REVIEWERS' COMMENTS

Reviewer #1 (Remarks to the Author):

The manuscript is well written and organized. I found that it is easy to follow for an specialized on magnetotellurics. I would like to suggest some changes/additions in order to increase its interest for non-specialists.

Please see the attached document for the comments

Reviewer #2 (Remarks to the Author):

Hill et al. present an electrical resistivity image from 129-station array of magnetotelluric data collected around Mount Erebus, Antarctica. The results show a rather continuous vertically oriented conductive body that extends from mantle depths to just 1 km beneath the surface. The authors describe the conductor's continuity as evidence for the lack of viscous stalling in CO₂-dominated alkali-rich magmas, which contrasts with the behavior of magma in subduction zone arc settings where viscous water-rich silicate melts stall at larger crustal depths of ~5 km. The authors also use the geometry of the conductor, which transitions to a more horizontally aligned trajectory at around 15 km depth, to argue for structural control on magma migration pathway and fault-valve behavior.

This is a compelling study that provides novel insight into the geometry of magma in a CO₂-dominated volcanic system. The manuscript is well organized and clearly written, the figures are nice, the discussion is logically presented, the analysis is thorough, and the technical details are well documented. It is suitable for publication in this journal. Honestly, I struggled to find weaknesses to complain about, hence I only have a few minor comments for the authors to consider.

Line 219: I think the authors meant 0.1–0.3 ohm-m?

Line 259: Can the authors elaborate on the length of seismic survey history? That would help to put into context what timespans are being referred to by a longer recurrence interval. I would have thought that fault-valve control and fluids at near-lithostatic pressures would be associated with high levels of (micro)seismicity.

Fig S4: It would be better to show the apparent resistivity curve of Rx91-Lo normalized by Rx91-Hi (or vice versa), as well as the difference of the phase curves. Given the scales, it is hard to gauge the similarity precisely, though qualitatively they are indeed similar. The same suggestion applies to the plots of phase tensor ellipses (Fig S14) and apparent resistivity from invariant tensor impedance (Fig S15). I struggled to compare the data to the models, which could be resolved if the differences are plotted in a panel alongside the absolute values.

Fig S16: The data-to-model response comparisons are useful. However, the two sites shown here are both good fitting examples. It would be worthwhile to get additional insight as to which period band the models struggle to fit the data for the relatively high RMS sites, many of which cluster around Mt Erebus summit.

Reviewer #3 (Remarks to the Author):

This is a polished, well-written, and well-organized manuscript describing exciting results from recent geophysical imaging of the CO₂-rich magmatic system of Earth's most continuously active phonolitic magmatic system, Erebus volcano, from conduit to mantle depths. The paper reports several discoveries of

note, including: 1) A continuous low-resistivity (melt-associated) structure from vent to upper mantle (~100 km) that is geometrically distinct from the more discontinuous and crust-segregated structural MT images seen at H₂O-rich systems in similar-style experiments (e.g., in the US Cascade Range; Hill et al., Nat. Geosci., 2009 [same first author and general methodology]). 2) The presence of an offset in this low-conductivity structure that they attribute to crustal scale displacement, and which may create tectonic control on mantle-surface connectivity; 3) The extension of low resistivity structure to very shallow depths hypothesized to be related to low water content [the alluded to lack of “volatile exsolution stalling” might explain the unusual long-standing open conduit/lava lake conditions at this volcano; which is worth mentioning in this context; e.g., in the last paragraph] . The inversion and data handling methodology are state-of-the-art (and are well documented in the supplemental materials; although more information on the regularization of the model and implications for resolving kernel width with depth might have been included). Notably, the depth restriction resolution test in the main paper [Figure 4] that demonstrated to my satisfaction that the deep structure is indeed required by the data. The paper is well referenced and well informed by the prior research in imaging, volcanology, and other studies on Erebus volcano. This large-scale image complements prior high-frequency seismic inversion for which resolution only extended to the uppermost few km of the volcano and very valuably highlights deeper structure that might be seismically imaged using longer-wavelength (scattered teleseismic earthquake) wavefields and/or indeed be better resolved via (the supplementary section suggestion of) follow-up MT measurements over the ice-covered Erebus Bay region in complementary future work that would further inform knowledge of the deep magmatic system.

Reviewer #1 (Remarks to the Author):

The manuscript is well written and organized. I found that it is easy to follow for an specialized on magnetotellurics. I would like to suggest some changes/additions in order to increase its interest for non-specialists.

Please see the attached document for the comments

We thank the reviewer for efforts to improve our ms. We have added to the text a number of the suggestions and attempted to explain our approach to model testing, which we believe is appropriate. Some noted aspects of presentation have the consensus of the author group, and did not raise concerns with the other reviewers.

Line 95: maximum spatial distance between MT sites, period range

We have modified the text to read "The MT data interpreted here provide structural resolution superior to seismic studies to date by imaging the electrical resistivity of the Erebus system continuously from the near surface to a depth of tens of kilometres. This is possible due to the very broad bandwidth of the MT signals (~0.005 – 900 s wave period) and the relatively dense site spacing (1 – 5 km) covering the entire Ross Island."

Line 104: I guess that you mean the Phase Tensor and that the bold meant that is a tensor. I think that the phase tensor description could be improved

*We have updated the text so that impedance phase **tensor** (refs) is specified and note that Φ_2 is the geometric mean of the Φ_{min} and Φ_{max} .*

Line 112: Please give a range of depths for this period.

We prefer to constrain the depth range through the quantitative tests presented subsequently. Stating a depth range here assumes a subsurface resistivity, which as will be seen is quite heterogeneous.

Line 113: Do you mean measured

We have added the adjective "observed".

Line 119: I think that it is difficult even for a specialist to draw so many information about the electrical resistivity distribution just with a component of the phase tensor at 1 period

We believe we are conservative about the information there. We simply list a few locations and note whether the phase ellipse Φ_2 values are above or below 45 deg. As is standard with phase, such relative colors imply whether the signals are penetrating decreasing or increasing bulk resistivity. Moreover, it is difficult to dispute that a large-scale alignment of phase ellipses implies some sort of structural direction. Additional phase tensor plots are provided in the supplementary material.

Line 120: Introduce the meaning of induction arrows. be aware that later you mention them as tipper.

We have defined induction arrows in the Methods and Supplementary Material and have noted that in the text.

Line 120: highlighted 'to'

We have put a comma after “summit crater” to distinguish this clause.

Line 137: First time introduced, before you were talking about the phase tensor

We have defined $E=ZH$ in the Methods and Supplementary Material and have noted that in the text.

Line 139: Induction arrows in the previous page

We have updated the terminology to induction arrows or vertical magnetic transfer function throughout.

Line 144: It is misleading this app. resistivity. Please compare the App res. and Phases of the 4 Impedance tensor components (measured and model responses). It is very difficult to check the quality of your model fitting with the graphics that you use

We don't see the justification for that statement. The apparent resistivity shown is based upon an impedance rotational invariant, which is robust to noise. It is clear from the plot that the impedance norm has a wide variation over the survey area and that our model fits it precisely with reference to the detailed color scale. Similarly the invariant phase tensor quantity is wide-ranging and fit. More specifically, the quality of a model should be judged by tests particular to its features that are relevant to the primary tectonic questions being posed or structures resolved. This we have focused on in the following subsections. We don't believe plotting a large volume of apparent resistivity and phase curves is insightful, and in fact would be disorienting.

Line 168: It will help to label these structures in the figures

We have added a label for Mts Terror and Terra Nova to the figure.

Line 177: In the supplementary material you only show two test cutting the model at 7 and 23 km depth. It will be good to 'cut' the model at deeper depths to see the resolution you have on the depth of the low resistivity structure.

This model test has a very specific purpose. It is to isolate the depth range of structure that is causing the high Φ_2 values over the southwestern portion of the survey area. This is the area that contains the youngest phonolite lavas. In order to see whether these young lavas have a specific connection to structure at depth, or alternatively that the Φ_2 anomaly includes the effects of structure at depth, we systematically truncated the model from below to see if the area of warm phase ellipses attenuated or disappeared. It did not. Thus, any deep structural connection to the young flows in this area is too subtle to be resolved if it exists at all.

Line 182: You can compare the model responses of both models to see which component of the impedance tensor and at which period there are differences between the two models.

We did not pursue this approach because the visual expression of the difference in model responses is subtle and uneven over the pertinent western portion of the survey area. We don't think the significance of different models will be well served by a large volume of sounding plots. The more rigorous test is the one that we presented, which is to see if structure can be confined to the crust and still provide as good a fit. This is done by taking the inverse-optimal model that tried to confine it so, and inverting its response using a noise structure identical to that of the observations. This test seems to demand that structure extends subvertically into the upper mantle.

Line 208: maybe because when you compare phase tensor values you are doping and 'average' of the four components. Checking the components individually app. resistivity and phases may help to determine where in the data are the 'precise characteristics of the observations leading to the upper mantle low resistivity'

Strictly from the standpoint of phase tensor display, the ellipses in Figure S21 do show that upper mantle structure does affect the response at several 10s of seconds (i.e., 86 s). In the constrained equivalent inversion of Figure 4, the differences in fit relative to the unconstrained inversion of Figures 2 and 3 are more subtle than would show clearly in a handful of individual plotted response curves in this case. We have looked at finite bands of nRMS but the change in fit from a visual standpoint is not better focused than in Figure S22. The fact that misfit changes are not smooth across sites suggests there are also local second order minima factoring on top of overall fit change. We believe the model test speaks for itself and little is to be gained from parsing the data effects further.

Line 211: In summary, I think that your model is good but I do not like how do you present the model testing.

We expect preferences in individual presentation approaches to vary, but our model testing is appropriately rigorous and convincing with respect to the major volcanological processes described.

Line 618: Circles : MT stations

We have noted this in the caption.

Line 749: Why do you choose this invariant. It is affected by galvanic distortion?
It is a bit confusing using several rotational invariants in the same paper

Only two invariants are used in the paper. The phase tensor invariant Φ_2 is one of those and it sees extremely wide use. Thus it is an appropriate one to display the presence of major structures relatively immune from galvanic distortion. The other is a popular impedance rotational invariant. All impedance invariants in principle may suffer from galvanic distortion. However, it is necessary to consider functions of the amplitude of the impedance (e.g. this one, and their derived apparent resistivities) because intrinsic depth and resistivity of structures derived from phase alone are highly if not completely non-unique. The invariant we use has the advantage of not involving impedance element products, where if one element is noisy then the invariant will be similarly noisy.

Line 817: change to induction arrows?

We no longer use the MT jargon 'tipper' and refer to either the vertical magnetic field transfer function or induction arrows throughout as defined at initial usage.

Line 839/840: Induction arrows?

Ditto.

Line 862: More tests with deeper truncations will be helpful to determine which components and periods of the impedance tensor are requested for the deeper structures

We have addressed the specific purpose of this test above.

Line 887: Couldn't find this reference (highlighted)

We have strived to place all the pertinent amplifier information in the Supplementary Methods, and the designer J Stodt is a coauthor on the paper. Thus we have removed the reference.

Reviewer #2 (Remarks to the Author):

Hill et al. present an electrical resistivity image from 129-station array of magnetotelluric data collected around Mount Erebus, Antarctica. The results show a rather continuous vertically oriented conductive body that extends from mantle depths to just 1 km beneath the surface. The authors describe the conductor's continuity as evidence for the lack of viscous stalling in CO₂-dominated alkali-rich magmas, which contrasts with the behavior of magma in subduction zone arc settings where viscous water-rich silicate melts stall at larger crustal depths of ~5 km. The authors also use the geometry of the conductor, which transitions to a more horizontally aligned trajectory at around 15 km depth, to argue for structural control on magma migration pathway and fault-valve behavior.

This is a compelling study that provides novel insight into the geometry of magma in a CO₂-dominated volcanic system. The manuscript is well organized and clearly written, the figures are nice, the discussion is logically presented, the analysis is thorough, and the technical details are well documented. It is suitable for publication in this journal. Honestly, I struggled to find weaknesses to complain about, hence I only have a few minor comments for the authors to consider.

We thank the reviewer for efforts to improve our ms. We have modified the text in places and explain our approach in the following response.

Line 219: I think the authors meant 0.1–0.3 ohm-m?

Actually, we did mean 1 – 0.3 as written. That is because the values are written in reference to the subsequently listed range of water content of 0 – 3 wt %. Melt resistivity falls as water content rises.

Line 259: Can the authors elaborate on the length of seismic survey history? That would help to put into context what timespans are being referred to by a longer recurrence interval. I would have thought that fault-valve control and fluids at near-lithostatic pressures would be associated with high levels of (micro)seismicity.

Thank you for this question, about which we have consulted others working on seismicity at Erebus. As reviewed by Rowe et al (2000), which we now cite, the first seismometers for volcanological surveying on Erebus were installed in 1973. Since then (until 2000 at least), seismic activity has been dominated by Strombolian eruptions with infrequent tremor, but no crustal earthquakes have been reported. It is unclear to us what a fault-valve related occurrence interval should be, and with the high temperatures inferred at only moderate depths below Erebus much of the valving activity may be close to ductile.

Fig S4: It would be better to show the apparent resistivity curve of Rx91-Lo normalized by Rx91-Hi (or vice versa), as well as the difference of the phase curves. Given the scales, it is hard to gauge the similarity precisely, though qualitatively they are indeed similar. The same suggestion applies to the plots of phase tensor ellipses (Fig S14) and apparent resistivity from invariant tensor impedance (Fig S15). I struggled to compare the data to the models, which could be resolved if the differences are plotted in a panel alongside the absolute values.

We have added a third panel to the plot overlaying the Lo and Hi energy responses and have plotted a grey bar representing the magnitude of the error floors used in the inversion, which is larger than the

difference between the hi and lo energy period responses.

Fig S16: The data-to-model response comparisons are useful. However, the two sites shown here are both good fitting examples. It would be worthwhile to get additional insight as to which period band the models struggle to fit the data for the relatively high RMS sites, many of which cluster around Mt Erebus summit.

The two sites displayed were not chosen specifically to show model fit, but rather to contrast first order response behavior east of the summit with that to its west. We are highlighting the relatively increasing apparent resistivities and lower phases as period increases on the east side with generally more decreasing apparent resistivities and higher phases at the longer periods on the west side. This signifies the greater amount of deep low resistivity to the west of the summit which we have imaged with the inversion. We have noted in the supplementary text now that the summit site misfits tend to be at the long periods. We do not know the cause, although this is not uncommon in MT soundings. Speculatively, perhaps MT sensor motion from Strombolian or tremor activity that is not completely removed by the referencing is at play.

Reviewer #3 (Remarks to the Author):

This is a polished, well-written, and well-organized manuscript describing exciting results from recent geophysical imaging of the CO₂-rich magmatic system of Earth's most continuously active phonolitic magmatic system, Erebus volcano, from conduit to mantle depths. The paper reports several discoveries of note, including: 1) A continuous low-resistivity (melt-associated) structure from vent to upper mantle (~100 km) that is geometrically distinct from the more discontinuous and crust-segregated structural MT images seen at H₂O-rich systems in similar-style experiments (e.g., in the US Cascade Range; Hill et al., Nat. Geosci., 2009 [same first author and general methodology]). 2) The presence of an offset in this low-conductivity structure that they attribute to crustal scale displacement, and which may create tectonic control on mantle-surface connectivity; 3) The extension of low resistivity structure to very shallow depths hypothesized to be related to low water content [the alluded to lack of "volatile exsolution stalling" might explain the unusual long-standing open conduit/lava lake conditions at this volcano; which is worth mentioning in this context; e.g., in the last paragraph]. The inversion and data handling methodology are state-of-the-art (and are well documented in the supplemental materials; although more information on the regularization of the model and implications for resolving kernel width with depth might have been included). Notably, the depth restriction resolution test in the main paper [Figure 4] that demonstrated to my satisfaction that the deep structure is indeed required by the data. The paper is well referenced and well informed by the prior research in imaging, volcanology, and other studies on Erebus volcano. This large scale image complements prior high-frequency seismic inversion for which resolution only extended to the uppermost few km of the volcano and very valuably highlights deeper structure that might be seismically imaged using longer-wavelength (scattered teleseismic earthquake) wavefields and/or indeed be better resolved via (the supplementary section suggestion of) follow-up MT measurements over the ice-covered Erebus Bay region in complementary future work that would further inform knowledge of the deep magmatic system.

We thank the reviewer for their supportive remarks. We think it is difficult to argue for anything more detailed than a substantial presence of low resistivity, sub-vertical structure extending into the upper mantle. Frankly, we were pleased that the test in Figure 4 was as affirming as is. Without being critical, we note there is a substantial range in petrological models for the source depths of the parental basanite magma, and thus being able to place structure in the upper mantle is a significant step forward. We have noted now in the main text that the shallow projection of low resistivity is in keeping with the persistent lava lake and that additional data aperture especially toward the southwest could improve resolution to depth.